# Gradient-based Optimization of Dataset Mixtures

## Abstract

Modern state-of-the-art machine learning models are often trained using a combination of heterogeneous data sources. However, the utility of different data sources as support for learning some target tasks is often not equivalent, motivating the need for automated methods of optimizing the relative contribution of each data source to the model. In this work, we propose a dataset optimization strategy that slices a normal model training step into a series of data source-specific updates and splices them back together in an optimal manner with respect to the loss on some target task dataset. We demonstrate the effectiveness of our algorithm across different scenarios and domains, including classification problems for vision models and for next-token prediction tasks in the language domain.

## 1 Introduction

Many state-of-the-art machine learning models are trained on mixtures of data from multiple sources. For example, The Pile (Gao et al., 2020), a commonly used large-scale language dataset, is composed of data from 22 different sources, including text from ArXiv, GitHub and Wikipedia. A natural question follows: How does each subset impact model performance, and how should the mixture be reweighted to optimize downstream scores? The importance of optimizing the data mixture is further pronounced when model users have specific downstream tasks in mind.

Many prior works investigated different strategies to automate the optimization process of data mixture coefficients (Albalak et al., 2023; Liu et al., 2024; Ge et al., 2024; Thrush et al., 2024; Zhao et al., 2024; Shimabucoro et al., 2024). Such methods tends to rely on heuristics or empirically observed correlations between training data properties and downstream model performance.

In this paper, we construct an optimizer that directly minimizes loss on a downstream task using gradient-based optimization. Central to our approach is the concept of a *data mixture gradient* – the derivative of the downstream validation loss with respect to the dataset mixture weights used during training. Computing the mixture gradient is unfortunately intractable due to the complex dependency between the mixture coefficient and the model parameters. Therefore, we develop a theoretically justified approximation of the mixture gradient that breaks the aforementioned dependency, allowing tractable optimization for the mixture coefficients.

We propose a two-stage approach that optimizes model parameters and mixture coefficients separately in each stage (see Figure 1). Assume we are optimizing a mixture over $N$ subsets.

- In Stage-I optimization, we optimize the model using some initial mixture coefficients for $N$ data subsets. As the model trains, we accumulate the gradient contributions from each of the $N$ data subsets into $N$ separate buffers.

- In Stage-II optimization, we aim to optimize the mixture coefficients to improve the outcome of training, as measured on a validation set. Using the $N$ individual gradient contributions stored during Stage-I, the final model parameters can be written as the parameters at initialization, plus a linear combination of the $N$ accumulated gradients in these buffers. We then optimize the $N$ coefficients in this linear combination to find a synthetic iterate that minimizes the validation loss. This simple optimization problem serves as a proxy for re-weighting the data mixture.

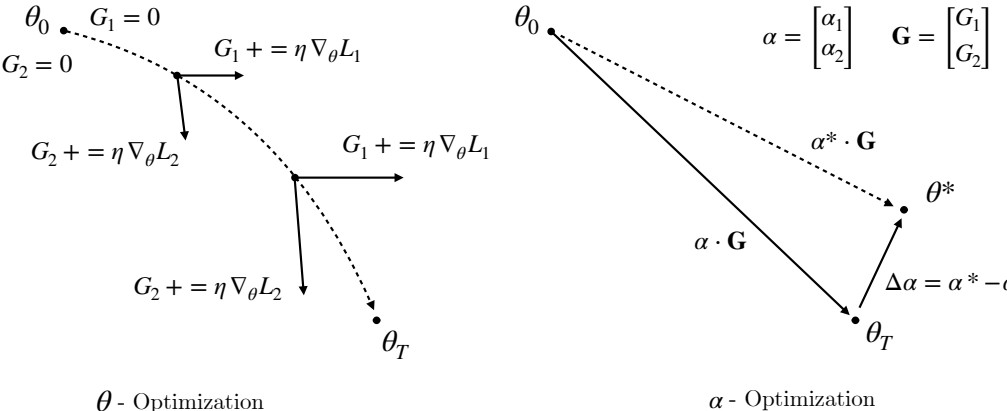

Figure 1: An illustration of our proposed two-stage algorithm to optimize dataset mixture coefficients $\boldsymbol{\alpha}$ for a given task. In the first stage (left figure), we train the model to obtain $\boldsymbol{\theta}_T$ and accumulate subset-specific gradient $\nabla_{\boldsymbol{\theta}} L_i$ to $G_i$. In the second stage, we represent the trained model as $\boldsymbol{\theta}_T = \boldsymbol{\theta}_0 + \boldsymbol{\alpha} \cdot \boldsymbol{G}$, and optimize $\boldsymbol{\alpha}$ while keeping $\boldsymbol{G}$ fixed. The new coefficients $\boldsymbol{\alpha}^*$ is used to produce a better model $\boldsymbol{\theta}^* = \boldsymbol{\theta}_T + \Delta\boldsymbol{\alpha} \cdot \boldsymbol{G}$.

We empirically demonstrate that Stage-II model obtained using linear combination of $N$ gradients weighted by the optimized coefficients indeed outperforms the Stage-I model. We further propose a checkpointing variant of our algorithm that is more friendly to large-scale datasets, and a multi-episode extension that performs the two-stage optimization in multiple rounds for better performance.

We present an interpretable dataset remixing experiments on DyckGrammer (Schützenberger, 1963; Yao et al., 2021; Wen et al., 2023), a synthetic language generation task, to visualize the effect of our algorithm. We further conduct in-depth experiments on carefully constructed datasets based on CIFAR-10 (Krizhevsky et al., 2009) to validate the effectiveness of our algorithm and its different variants. Finally, we apply our algorithm to an object detection task, where we experiment with DETR (Carion et al., 2020) trained on the COCO dataset (Lin et al., 2014). We show that our algorithm improves test-time performance by reweighting the classes in the dataset.

## 2 OPTIMIZING DATASET MIXTURE THROUGH GRADIENT REMIXING

Consider a collection of datasets denoted as $\mathcal{D} = \{D_i\}_{i=1}^N$, where each dataset $D_i = \{(x_j, y_j)\}_{j=1}^{m_i}$ consists of $m_i$ training examples. Here, we have omitted the dataset-specific index on the training examples $(x_j, y_j)$ for notational simplicity. In practice, the dataset collection $\mathcal{D}$ may consist of individual datasets from different sources, such as arXiv, code, or math. Alternatively, it can also be constructed by partitioning a single dataset into multiple subsets if the partitioning criteria is known.

The standard approach of training a model $f(:, \boldsymbol{\theta})$ with parameters $\boldsymbol{\theta}$ (alternatively denoted by $f_{\boldsymbol{\theta}}$) involves minimizing a loss that is a mixture of data from various sources.

$$L(\boldsymbol{\theta}; \boldsymbol{\alpha}) = \sum_{i=1}^N \alpha_i L_i(\boldsymbol{\theta}) = \boldsymbol{\alpha} \cdot L(\boldsymbol{\theta}). \qquad (1)$$

Here, $L_i(\boldsymbol{\theta}) = \frac{1}{|D_i|} \sum_{x,y \in D_i} \ell(f_{\boldsymbol{\theta}}(x), y)$ is the loss associated with dataset $D_i$, $L(\boldsymbol{\theta})$ is a vector-valued function containing each of the individual losses and $\boldsymbol{\alpha}$ is a vector of mixture coefficients.

In many practical settings, the downstream performance of a model depends strongly on the choice of the mixture coefficients. Our goal is to optimize the choice of $\boldsymbol{\alpha}$ to achieve the lowest possible loss on a test set. A dataset mixture can be evaluated by first training on the rebalanced data, and then evaluating the resulting model on a validation set. More formally, we train a model on the data mixture using gradient descent:

$$\boldsymbol{\theta}_t = \boldsymbol{\theta}_{t-1} - \eta_{t-1}L'(\boldsymbol{\theta}_{t-1};\boldsymbol{\alpha}) \qquad \text{for } t = 1, 2 \ldots T, \tag{2}$$

where $\{\eta_t\}$ is a sequence of learning rates and $L'(\boldsymbol{\theta}_{t-1};\boldsymbol{\alpha})$ is the gradient. Then, we evaluate the validation loss $V(\boldsymbol{\theta}_T)$. The goal of dataset mixology is to optimize $\boldsymbol{\alpha}$ to achieve low downstream loss. In our proposed approach, we first train the model parameters $\boldsymbol{\theta}$, and optimize $\boldsymbol{\alpha}$ to minimize the downstream loss. A key feature of our approach is that we approximate the *mixture gradient* – the derivative of the downstream loss with respect to the upstream mixture coefficients – thus enabling efficient and effective optimization of $\boldsymbol{\alpha}$.

## 2.1 A DIFFERENTIABLE FORMULATION OF THE PROBLEM

We would like an efficient way to optimize $\boldsymbol{\alpha}$ to minimize the validation loss $V(\boldsymbol{\theta}_T)$. Note that $\boldsymbol{\theta}_T$ implicity depends on $\boldsymbol{\alpha}$, although this dependence is fairly complex. We will make a simplifying assumption to remove this complexity.

The simple idea behind our proposed method is to apply Equation (2) recursively to write

$$\boldsymbol{\theta}_T = \boldsymbol{\theta}_0 - \sum_{t=1}^{T-1} \eta_t \boldsymbol{\alpha} \cdot L'(\boldsymbol{\theta}_t)$$

$$= \boldsymbol{\theta}_0 - \alpha_1 \sum_{t=1}^{T-1} \eta_t L_1'(\boldsymbol{\theta}_t) - \alpha_2 \sum_{t=1}^{T-1} \eta_t L_2'(\boldsymbol{\theta}_t) \cdots - \alpha_N \sum_{t=1}^{T-1} \eta_t L_N'(\boldsymbol{\theta}_t)$$

$$= \boldsymbol{\theta}_0 - [\alpha_1 G_1 - \alpha_2 G_2 \cdots - \alpha_N G_n] = \boldsymbol{\theta}_0 - \boldsymbol{\alpha} \cdot \boldsymbol{G}. \tag{3}$$

Where the gradient contribution of the $i$th dataset is $G_i = \sum_{t=1}^{T-1} \eta_t L_i'(\boldsymbol{\theta}_t)$. This expansion writes the final iterate $\boldsymbol{\theta}_T$ as $\boldsymbol{\theta}_0$ plus all the individual gradient contributions from each dataset. Even though $\boldsymbol{\alpha}$ does not appear in the formula for $G_i$, these gradient contributions depend implicitly on $\boldsymbol{\alpha}$ because a different choice of $\boldsymbol{\alpha}$ would result in a different trajectory of iterates.

Our proposed method uses a simplification to make dataset optimization tractable: we treat each $G_i$ like a constant. After running the training loop to find $\boldsymbol{\theta}_T$, we then formulate the dataset optimization problem

$$\min_{\boldsymbol{\alpha}} V(\boldsymbol{\theta}_T) \approx V(\boldsymbol{\theta}_0 - \boldsymbol{\alpha} \cdot \boldsymbol{G}). \tag{4}$$

We can easily minimize the objective using gradient-based optimization. The gradient of $V$ with respect to $\boldsymbol{\alpha}$ can be computed using autograd, or using the formula

$$\frac{\partial}{\partial \boldsymbol{\alpha}} V(\boldsymbol{\theta}_T) \approx \frac{\partial}{\partial \boldsymbol{\theta}} V(\boldsymbol{\theta}_T) \frac{\partial}{\partial \boldsymbol{\alpha}} [\boldsymbol{\theta}_0 - \boldsymbol{\alpha} \cdot \boldsymbol{G}] = -V'(\boldsymbol{\theta}_T) \cdot G. \tag{5}$$

Note that we use the $\approx$ symbol to emphasize that this gradient treats $G$ as a constant and ignores its implicit dependence on $\boldsymbol{\alpha}$.

## 2.2 THEORETICAL JUSTIFICATION

The approximate mixture gradient in Equation (5) results from treating $\boldsymbol{G}$ like a constant. In this section, we show that our approximate gradient closely matches the true dataset mixture gradient when the learning rate $\eta$ is small.

**Theorem 1.** *Consider the downstream validation loss*

$$\nabla_{\boldsymbol{\alpha}} V(\boldsymbol{\theta}_T) := \frac{\partial}{\partial \boldsymbol{\alpha}} V(\boldsymbol{\theta}_T), \tag{6}$$

*where $\boldsymbol{\theta}_T$ is given by Equation* (2) *and implicity depends on $\boldsymbol{\alpha}$. We then have*

$$\frac{\partial}{\partial \boldsymbol{\alpha}} V(\boldsymbol{\theta}_T) = - \underbrace{V'(\boldsymbol{\theta}_T) \cdot G}_{\textit{first order term}} - V'(\boldsymbol{\theta}_T) \cdot \underbrace{\sum_{t=0}^{T-1} \eta_t \boldsymbol{\alpha} \cdot \frac{\partial}{\partial \boldsymbol{\alpha}} [L'(\boldsymbol{\theta}_t)]}_{\textit{higher order term}}. \tag{7}$$

We provide the proof in Appendix C.

Our approximate mixture gradient in Equation (5) matches the first-order term in the expansion for the true dataset gradient, while neglecting the terms involving higher-order derivatives. For small, constant learning rate $\eta$, the first order term is $O(\eta)$ because of the factor of $\eta$ that is included in the definition of $G$. Meanwhile, the term $\frac{\partial}{\partial \boldsymbol{\alpha}}[L'(\boldsymbol{\theta}_t)] = \frac{\partial}{\partial \boldsymbol{\alpha}}[L'(\boldsymbol{\theta}_0 - \eta \sum_{t=0}^{t-1} \boldsymbol{\alpha} \cdot L'(\boldsymbol{\theta}_t))] \approx O(\eta)$. Combining this observation with the additional factor of $\eta$ in front of the higher order term, we see that this term has magnitude $O(\eta^2)$, making our approximation accurate for small $\eta$.

Note that we often train with as large a learning rate as possible, and some of the mixed partial derivatives in the neglected higher order term may be large. In practice, we do not expect the higher order terms in equation 7 to be vanishingly small. Nonetheless, we find that our simple first-order approximation of the mixture gradient is good enough to succeed in many situations.

## 2.3 OUR METHOD

We discuss our method in detail in this section. The overall training process can be viewed as an alternative optimization method that optimizes model parameters $\boldsymbol{\theta}$ and mixture coefficients $\boldsymbol{\alpha}$ in two stages:

**Stage I:** ($\boldsymbol{\theta}$-optimization) We train our model parameters $\boldsymbol{\theta}$ on the data mix given by Equation (1), while keeping the mixture coefficients $\boldsymbol{\alpha}$ fixed. As training proceeds, we store the cumulative gradient from $D_i$ in vector $G_i = \sum_{t=0}^{T-1} \eta_t L_i'(\boldsymbol{\theta})$, allowing us to separate out the contribution that each dataset makes to the final iterate. Note that while we motivate our approach based on gradient descent, our algorithm is not restrictive on the choice of model optimizers (as shown in Section 3.2).

Also note that both the loss $L_i(\boldsymbol{\theta}_t; \boldsymbol{\alpha})$ and the gradient $L_i'(\boldsymbol{\theta}_t; \boldsymbol{\alpha})$ are computed using a mini-batch sampled from $D_i$. This mini-batch changes on each step $t$, although our gradient notation omits this dependence to avoid clutter.

**Stage II:** ($\boldsymbol{\alpha}$-optimization) In the second stage, we keep the model parameters $\boldsymbol{\theta}$ fixed and optimize the mixture coefficients starting from $\boldsymbol{\alpha}_0$ by minimizing the validation loss $V(\boldsymbol{\theta}_T)$ as defined in Equation (6). This simple procedure remixes the gradient information from individual datasets and results in an updated model with parameters $\boldsymbol{\theta}^*$ that achieves better validation performance.

Rather than rely directly on Equation (6), we make the change of variables $\boldsymbol{\beta} := \boldsymbol{\alpha} - \boldsymbol{\alpha}_0$. This converts Equation (6) to the equivalent form

$$\min_{\boldsymbol{\beta}} V(\boldsymbol{\theta}_T - \boldsymbol{\beta} \cdot \boldsymbol{G}). \tag{8}$$

We refer to optimizing Equation (8) as the 'wiggle' method as it corresponds to letting the final iterate wiggle around $\boldsymbol{\theta}_T$ rather than starting optimization from the far-away iterate $\boldsymbol{\theta}_0$. The optimized $\boldsymbol{\beta}^*$ can be used to compute the mixture coefficients $\boldsymbol{\alpha}^* = \boldsymbol{\alpha}_0 + \boldsymbol{\beta}^*$ if further training is required.

Why use the wiggle method? Both Equation (6) and Equation (8) are the same when the gradient contributions $G$ are calculated exactly. In situations where $G$ is inexact (e.g., because of low precision training), it is better to use Equation (8) as it always starts optimizing from the iterate $\boldsymbol{\theta}_T$.

For various reasons such as interpretability of $\boldsymbol{\alpha}$, it is sometime preferred to have the mixture contributions sum up to one. In this case, we use a softmax function $\phi(.)$ to normalize $\boldsymbol{\alpha}$ values during training. We provide the details of this formalism in Appendix D.

## 2.4 EXTENSIONS

**Gradient Approximation for Large Scale Training:** The gradient storage step during Stage I requires keeping track of $N$ copies of model gradients, making the memory requirement infeasible at a large scale if these copies are simply stored in memory. One can also store the model gradients in disks, but this incurs infeasible computation overhead as each gradient copy must be moved between memory and disk in each iteration whenever gradient accumulation happens. To overcome this, we store $P$ checkpoints of model $\boldsymbol{\theta}$ evenly spanned across training steps during Stage I, and

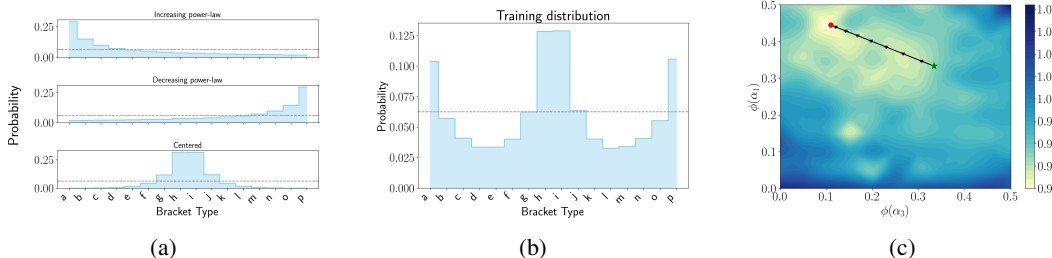

(a)  (b)  (c)

Figure 2: (a) The bracket distributions of different Dyck grammar training datasets, (b) the combined bracket distribution of the training datasets, and (c) the landscape of the target dataset loss in the mixture coefficient space. Lighter color corresponds to a lower loss value on the target distribution. The optimization trajectory of $\boldsymbol{\alpha}$ is shown as black lines with arrows, with the green star and red circle marking the initialization and the end of optimization.

approximate the gradient information $G$ as

$$G_i \approx \sum_{p=1}^{P} \eta_p L'_i(\boldsymbol{\theta}_p; \boldsymbol{\alpha}_p), \tag{9}$$

where $p$ indexes the checkpoint and $\eta_p$ represents the corresponding learning rate at the training step $\boldsymbol{\theta}_p$ is stored. To further reduce the computation for large-scale datasets, we only calculate the gradient $g_i(\boldsymbol{\theta}_p)$ on a random subset of the full training dataset. The frequency to recompute $G_i$ using a new random training subset is a hyperparameter and can be set based on computational budget.

**Multi-episode Training:**  Here, we discuss how our method can be extended to run in multiple episodes, where each episode is a single run of Stage I and Stage II optimization. Naively, after finishing an episode $\mu$ to obtain $\boldsymbol{\theta}^*$ and $\boldsymbol{\alpha}^*$, one can simply set $\boldsymbol{\theta}_0^{(\mu+1)} = \boldsymbol{\theta}^*$ and $\boldsymbol{\alpha}_0^{(\mu+1)} = \boldsymbol{\alpha}^*$ for a new episode $\mu + 1$, and continue to run another two-stage optimization. However, during stage II, we normally only observe a minimal change in mixture coefficients $\boldsymbol{\alpha}$, particularly in the later episodes (as shown in Appendix B). Consequently, while the resulting $\boldsymbol{\theta}^*$ can already substantially improve the test-time performance over $\boldsymbol{\theta}_T$, the corresponding $\boldsymbol{\alpha}^*$ generally does not carry sufficient difference to be impactful in the next episode. On the other hand, despite the small magnitude, these changes still provide valuable information about the optimal direction for adjusting $\boldsymbol{\alpha}$. To leverage this information, we apply discrete updates to $\boldsymbol{\alpha}$ at the end of each episode based on the sign of the $\boldsymbol{\alpha}$ changes as follows

$$\boldsymbol{\alpha}_0^{(\mu+1)} = \boldsymbol{\alpha}_0^{(\mu)} + \gamma \, \mathrm{sgn}(\boldsymbol{\alpha}^* - \boldsymbol{\alpha}_0^{(\mu)}). \tag{10}$$

Here, $\gamma$ controls the magnitude of the update and represents the step size of optimization in the dataset space. This allows us to make more substantial updates to the mixture coefficients, potentially overcoming local optima and encouraging further exploration of the optimization landscape.

The above process is repeated until the validation loss converges, with each new episode building on the previous one. In Appendix B, we provide additional experiments that motivated different design choices of our algorithm.

## 3 EXPERIMENTAL RESULTS

### 3.1 DYCK GRAMMAR: A TOY EXAMPLE

To illustrate the inner workings of our algorithm, we present an interpretable experiment on the bounded Dyck grammar Schützenberger (1963); Yao et al. (2021); Wen et al. (2023). The Dyck grammar consists of balanced brackets of multiple types, constituting a formal language grammar. For two types of brackets $\{\}, ()$, an example string looks like: $\{()\{\}\}()\{\{\}\}$.

We consider Dyck grammar with 16 types of brackets and a maximum nested depth of 8. We construct three training datasets, each with a different distribution for the brackets —(1) an increasing

power-law, (2) a decreasing power-law, and (3) an exponentially decreasing distribution from the center. Figure 2(a, b) illustrates these distributions, along with the combined training dataset distribution with uniform dataset mixture, i.e., $\{\alpha_i = 1\}_{i=1}^{N=3}$.

For the target dataset, we consider a uniform distribution of the brackets. The goal of our algorithm is to determine the optimal dataset mixture coefficient $\{\alpha_i^*\}_{i=1}^{N=3}$ that minimizes the loss on the uniform distribution. Intuitively, we expect $\alpha_1$, $\alpha_2$ to increase and $\alpha_3$ to decrease. Specifically, larger values of $\alpha_1$ and $\alpha_2$ increase the probability of underrepresented brackets, while a smaller value of $\alpha_3$ reduces the probability of over-represented central brackets.

Figure 2c shows the loss landscape on target dataset with respect to normalized coefficient $\phi(\boldsymbol{\alpha})$, generated by running model optimization with $\boldsymbol{\alpha}_0$ initialized as the sets of coefficients whose normalization supports the whole landscape. In this landscape, the value of $\alpha_2$ is inferred by $\phi(\alpha_2) = 1 - \phi(\alpha_1) - \phi(\alpha_3)$. As expected, the target dataset loss is small for large values of $\phi(\alpha_1), \phi(\alpha_2)$ and small values of $\phi(\alpha_3)$.

Next, we show the effectiveness of our algorithm in finding the optimal data mixture coefficient in this loss landscape. We initialize $\boldsymbol{\alpha}_0$ with a uniform mixture (marked by a green star in Figure 2c), and run the multi-episode variant of our algorithm. As the training episode progresses, our algorithm gradually updates $\boldsymbol{\alpha}$ to reduce the loss on the target distribution and eventually converges close to the minimum, as marked by the red circle. Further discussion, including model and optimization details, are provided in Appendix A.1.

## 3.2 IMAGE CLASSIFICATION

We perform an in-depth study on our algorithm and its different extensions on image classification tasks using CIFAR-10 (Krizhevsky et al., 2009). We construct two different datasets based on different splitting of the original CIFAR-10.

- **Mislabeled CIFAR-10.** This dataset contains two training subsets. One is the ordinary CIFAR-10. We construct the other one by intentionally reassigning the target of each example in CIFAR-10 with an incorrect label. In this scenario, the optimal mixing coefficient is clearly 1 on the correctly-labeled dataset and 0 on the other.

- **Imbalance CIFAR-10.** For this dataset, we construct an imbalance training set from CIFAR-10 as described in (Cao et al., 2019) with a balance ratio of 10, and split training set into 10 subsets based on class labels. As the class distribution between the training set and the test set is different, model trained with uniform $\boldsymbol{\alpha}$ will typically result in unsatisfactory performance.

We perform different experiments on the aforementioned two datasets summarized as follows.

- We examine how our algorithm paris with different model optimizers.

- We validate and ablate our checkpointing extension for further use in larger scale experiments described in Section 3.3.

- We compare our algorithm with direct fine-tuning of the model on the validation set to understand how well our algorithm leverages the additional data.

- We examine our multi-episode extension and its performance improvement relative to the single-episode version.

Additional experimental details are discussed in appendix A.2.

**Single-episode: Different optimizers.** We experiment with SGD, SGD with momentum and weight decay (SGD-wdm), and Adam as our model optimizer. All three optimizers use the same learning rate of 0.1. SGD-wdm is set up with momentum 0.9 and weight decay 0.0005. Adam is setup with $(\beta_1, \beta_2) = (0.9, 0.999)$. We run a single episode of our algorithm on both mislabeled CIFAR-10 and imbalance CIFAR-10, and report the performance on test-set after Stage-I and Stage-II optimization in Table 1. From the results, we observe that for all three optimizers, our stage II optimization consistently discovers an $\boldsymbol{\alpha}^*$ that better remixes the $\boldsymbol{\theta}_T$ with gradient information $\boldsymbol{G}$ into $\boldsymbol{\theta}^*$ with significantly improved test-time performance. The results for SGD optimizer show that

| Model Optimizer | | Mislabeled CIFAR-10 | Imbalance CIFAR-10 |
|---|---|---|---|
| SGD | Stage I | 0.416 | 0.688 |
| | Stage II | **0.910** | **0.771** |
| SGD-wdm | Stage I | 0.568 | 0.725 |
| | Stage II | **0.918** | **0.834** |
| Adam | Stage I | 0.500 | 0.631 |
| | Stage II | **0.639** | **0.714** |

Table 1: Test-time accuracy for a single-episode run of our algorithm on mislabeled CIFAR-10 and imbalance CIFAR-10. Stage-I indicates the performance of model $\boldsymbol{\theta}_T$ after $\theta$-optimization. Stage-II indicates the performance of $\boldsymbol{\theta}^*$ after $\boldsymbol{\alpha}$-optimzation.

our simple approximation of the mixture gradient is effective. The results on SGD-wdm and Adam show that $\boldsymbol{G}$ still provides valuable information toward recreating a better model even if Equation (2) does not hold.

| | Checkpoint # | Random Example # | Mislabeled CIFAR-10 | Imbalance CIFAR-10 |
|---|---|---|---|---|
| Stage I | | | 0.568 | 0.725 |
| Exact | | | 0.918 | 0.834 |
| Stage II | 5 | 1000 | 0.745 | 0.746 |
| | | 5000 | 0.915 | 0.768 |
| | | 10000 | **0.923** | **0.787** |
| | 10 | 1000 | 0.886 | 0.738 |
| | | 5000 | 0.913 | 0.751 |
| | | 10000 | 0.859 | 0.765 |

Table 2: Test-time accuracy for a single-episode run of our checkpointing extension on mislabeled CIFAR-10 and imbalance CIFAR-10. We show the results on different combinations of checkpoint number $P$ and the amount of random training examples. "Exact" indicates the results obtained using our normal algorithm.

**Single-episode: Checkpointing.** In this experiment, we validate the checkpointing extension of our algorithm, where the gradient information is approximated in Stage II with $P$ checkpoints of model $\{\boldsymbol{\theta}_p\}_{p=1}^P$ recorded in different training steps. We perform an ablation study on the effect of the number of checkpoints $P$ and the number of random training examples used to approximate the gradient information $G$. For this experiment, we recompute the approximation of $\boldsymbol{G}$ with different random samples in each iteration. We summarize the results on mislabeled CIFAR-10 and imbalance CIFAR-10 in Table 2. The results indicate that the checkpointing variant of our algorithm also consistently discovers $\boldsymbol{\theta}^*$ that outperforms $\boldsymbol{\theta}_T$ on the evaluation set, and the best results are achieved with $P = 5$ and 10000 random training examples. While the best performance is not always comparable to the normal variant, our checkpointing extension is still an effective variant more suitable for large-scale datasets.

| Val. Size | | Mislabeled CIFAR-10 | Imbalance CIFAR-10 |
|---|---|---|---|
| | Stage I | 0.568 | 0.725 |
| 5000 | Fine-tune | **0.920** | **0.841** |
| | Stage II | 0.915 | 0.814 |
| 2500 | Fine-tune | **0.924** | **0.830** |
| | Stage II | 0.917 | 0.815 |
| 500 | Fine-tune | 0.906 | 0.820 |
| | Stage II | **0.917** | **0.830** |
| 250 | Fine-tune | 0.878 | 0.799 |
| | Stage II | **0.916** | **0.812** |

Table 3: Test-time accuracy for a single-episode run of our method versus simple fine-tuning on mislabeled CIFAR-10 and imbalance CIFAR-10.

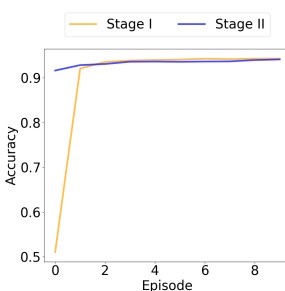 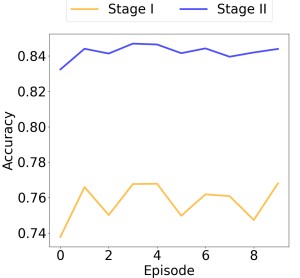 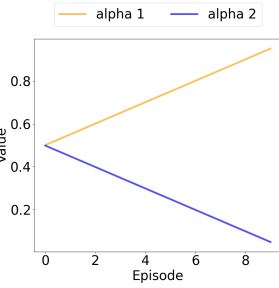

(a) Acc. for Mislabeled CIFAR    (b) Acc. for Imbalance CIFAR    (c) $\alpha$ curve for Mislabeled CIFAR

Figure 3: Multi-episode experiment on mislabeled CIFAR-10 and imbalance CIFAR-10. In Figure 3c, $\alpha_1$ is the weight of the correctly labeled dataset, and $\alpha_2$ is for the incorrectly labeled dataset.

**Fintuning vs Stage-II optimization.** In this experiment, we perform an ablation study to understand how effective our Stage-II optimization leverages the additional validation sets to improve the model. We compare our algorithm with a simple baseline where $\theta_T$ is directly fine-tuned on the validation set with the same optimizer configuration in Stage I except for the learning rate, which we set to $0.0001$. We summarize the results in Table 3. We observe that our Stage-II optimization outperforms simple fine-tuning when the size of the validation set is small, and the gap further increases when the validation set size gets smaller. This makes our algorithm more attractive empirically as a smaller validation set requires fewer resources to collect.

**Multi-episode Optimization with Qunatized Update** Finally, we examine the effectiveness of our algorithm when extended to multi-episode optimization through the quantized update rule as described in Equation (10). We perform 10-episode optimization and experiment with $\gamma \in \{0.1, 0.05, 0.01, 0.005, 0.001, 0.0001\}$ on both CIFAR-10 tasks, and illustrate the results with the best $\gamma$ in Figure 3. We also summarize the final Stage-II accuracy in Table 4. From Figure 3a and Figure 3b, our algorithm consistently outperforms the normally trained model $\theta_T^{(0)}$ after multi-episode optimization, and further improves upon the $\theta^*$ obtained by single-episode optimization as shown in Table 4. Our quantized update rule is also capable of guiding the trajectory of $\alpha$ toward the optimal data mixture, as shown in Figure 3c.

|  | SE | ME |
|---|---|---|
| Mislabeled | 0.918 | **0.941** |
| Imbalance | 0.834 | **0.851** |

Table 4: Final Stage II accuracy. SE: single-episode. ME: multi-episode.

### 3.3 OBJECT DETECTION

In this experiment, we test our algorithm on object detection models trained on COCO (Lin et al., 2014) datasets. We experiment with DETR Carion et al. (2020), an end-to-end object detection model based on transformers. Compared to previous experiments, COCO features complicated object detection tasks in a larger scale, and DETR is a much larger model that is computationally intensive to train. Each example in the COCO dataset can be represented as $(x, \{b_j, y_j\}_{j=1}^{m_x})$, where $m_x$ is the number of the bounding boxes associated with image $x$, and $(b_j, y_j)$ is the $j$-th bounding box and the corresponding class labels.

|  | Base | Ours. |
|---|---|---|
| Avg. AP | 41.9 | **42.4** |
| AP@50 | 62.3 | **62.5** |
| AP@75 | 44.1 | **44.9** |

Table 5: Performance of our algorithm applied to DETR trained on COCO.

We test the checkpointing variant of our algorithm in this scenario, where we separate the gradient information based on class label into a total of 91 buffers. We store 5 checkpoints during $\theta$-optimization stage. In Stage II, we sample new random training examples every 100 iteration and recompute $G_i$ accordingly. To determine a specific gradient approximation $G_i$ on a checkpoint $\theta_p$, we compute the model gradient for $\theta_p$ on each of the random examples, and sum over individual gradients whose corresponding image $x$ has at least one $y_j = i$.

We summarize the results of our algorithm in Table 5, where average AP is the average of AP whose IoU ranges from $0.5$ to $0.95$. The results show up to $0.8$ of improvement in average precision score. Note that unlike the previous scenarios, evaluation distribution and training distribution stays the same in this experiment. Furthermore, the training loss does not directly correspond to the evaluation metric, and therefore the performance improvement is significantly harder to achieve. Still, the results verify the usefulness of individual gradient information, and show that our algorithm can leverage such information to consistently improve the model with a better mix of gradient information.

## 4 RELATED WORKS

Here we briefly review related literature, which broadly fits into three categories.

**Dataset Mixture Optimization.** This line of works focuses on optimizing data mixture for a pre-trained model, particularly in language domain, and does not consider optimization for specific given tasks (Albalak et al., 2023; Liu et al., 2024; Ge et al., 2024; Thrush et al., 2024; Zhao et al., 2024; Shimabucoro et al., 2024). For example, Liu et al. (2024) constructs a linear model to predict the performance from mixture coefficients, where the data pair comes from training multiple small models with different coefficients, and obtain optimal coefficients by reverse optimization. Albalak et al. (2023) models the data mixture selection as a multi-armed bandit problem where a selection policy is used to sample training batches. A reward is estimated based on the losses for the batch, and the policy is updated accordingly. Contrary to these works, our paper focuses on obtaining optimal mixture coefficients and the corresponding model for a given target task.

**Data Selection via Influence Function/Attribution.** Broadly speaking, this line of works focuses on emphasizing the beneficial training examples for a given task during the training, where different strategies are proposed to gauge the benefit of a training example. For instance, Xie et al. (2023) computes the n-gram statistics from a reference dataset that has the same distribution as the target task, and calculates the importance score of each training example based on the statistics. They then factor in the importance score by performing importance reweighting during the model training. Pruthi et al. (2020) focuses on improving the scalability of influence function calculation, where they propose to approximate the influence using a subset of network layers and a number of saved layer checkpoints from a normal training run, and use the influence score for data selection. To handle a new target task, a user generally needs to fully repeat these algorithms, making adaptation between target tasks less efficient. On the other hand, our two-stage approach allows users to shift between target tasks on the fly without additional model training runs.

**Active Learning and Coreset.** Active learning algorithms naturally generate optimized selections of training data as a byproduct, and the coreset optimization problem Sener & Savarese (2018) is key to many related literature Sener & Savarese (2018); Mirzasoleiman et al. (2020); Xia et al. (2023); Coleman et al. (2019), where the optimization determines a fixed number of most informative examples out of the training dataset. However, the corset optimization depends on distance measurements between examples, which is known to be inaccurate in high-dimensional space. Furthermore, the scalability of active learning algorithms remains an open question.

## 5 CONCLUSION

In this paper, we propose an algorithm to optimize data mixture coefficients for a given target task. Our algorithm is a two-stage approach that alternates between optimization of model parameters $\theta$ and mixture coefficients $\alpha$. We develop an approximation for the mixture gradient backed by theoretical analysis, which allows us to efficiently optimize $\alpha$ in Stage II by keeping gradient information $G$ constant. The approximation also allows us to efficiently resemble a new model with improved test-time performance. We further propose a checkpointing extension that is computationally feasible for large-scale datasets, and the multi-episode extension which guides $\alpha$ to the optimal mixture coefficients and further improves the performance on the target task. We visually demonstrate the effectiveness of our algorithm in Dyck Grammer experiment, and validate different

variants of our algorithms in the CIFAR-10 experiment. We also show that our algorithm is effective for modern-scale datasets and models in the object detection experiment.

## REPRODUCIBILITY STATEMENT

We provide hyperparameters and other details of experiment settings of all our experiments in Appendix A as well as relevant sections. We also provide further detail and derivation to implement $\alpha$ normalization in Appendix D. The proof of Theorem 1 is detailed in Appendix C.

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

## A    EXPERIMENTAL DETAILS

### A.1    DYCK GRAMMAR

**Datasets:**    We consider Dyck grammar with $k = 16$ types of brackets and a maximum nested depth of $D = 8$, with a minimum length of 10 brackets and a maximum length of 50 brackets. We construct three training datasets with different bracket distributions, with $i$ indexing the brackets:

1. **Increasing power-law:** The bracket probability decreases with index as

$$P(i) \sim 1/i.$$

2. **Decreasing power-law:** The bracket probability increases with index as as
$$P(n) \sim 1/(k - i).$$

3. **Exponentially decreasing from the center:** The bracket probability decreases exponentially from the central bracket as
$$P(i) \sim \exp(-d),$$
   where $d$ is the distance from the central bracket.

Figures 2a and 2b shows these distributions, along with the combined training dataset distribution with equal contributions. For the target dataset, we consider the uniform distribution $P(i) = 1/k$ for the brackets. We generate $10^6$ tokens for each dataset, resulting in approximately $42,000$ examples. We use around $24,000$ examples for training and $8,000$ examples for both validation and test datasets.

**Transformer model:**    We use GPT-style Transformers Vaswani et al. (2017) with learnable positional encodings. The model consists of 4 layers, 4 attention heads, and an embedding dimension of $n_{\text{embd}} = 768$. For activation, we use GeLU activation. Notably, we do not use biases and do not employ weight tying.

**$\theta$-optimization:**    For Stage I, we use SGD optimizer with a learning rate $\eta = 0.3$ and a batch size of $B = 512$, without momentum. We use a linear warmup of 512 steps. Each $\theta$ optimization stage consists of $T = 1000$ training steps.

**$\alpha$-optimization:**    For the $\alpha$ stage, we employ GD optimizer with a learning rate $\eta_\alpha = 10^{-4}$ and, utilizing the entire validation dataset. We do not employ learning rate warmup during this stage. Each $\alpha$-optimization stage lasts for $T_\alpha = 100$ steps. For the quantized method, we use $\gamma = 0.1$.

**Dataset Landscape Generation:**    To generate the dataset landscape, we train the Transformer models for $T = 5000$ steps, with $\alpha$'s fixed throughout training; no $\alpha$ optimization was performed. All other hyperparameters are the same as described above.

### A.2    CIFAR-10

For all of our CIFAR-10 experiments, unless otherwise noted, we use ResNet-18 (He et al., 2016) as our model architecture. For all stage I optimization, we train the model $\theta$ for 200 epochs, using batch size of 100 and SGD optimizer with learning rate of 0.1, momentum of 0.9 and weight decay of 0.0005. For all stage II optimization, we optimize $\alpha$ for 2000 iterations, using default Adam optimizer with learning rate 0.0001. We partition a subset of 500 examples from CIFAR-10 test set as the validation set, and use the rest for evaluation. Our experiment starts with uniform dataset mixture coefficients.

## B    ADDITIONAL RESULTS

### B.1    DYCK GRAMMAR

In this section, we provide additional results that motivate the quantized update method. We consider the Dyck Grammar setup discussed in Section 3.1. Figures 4 and 5 compare the test loss and $\alpha$

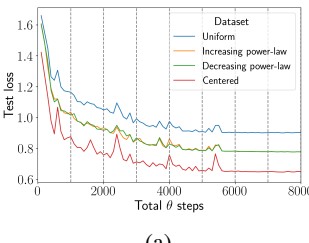 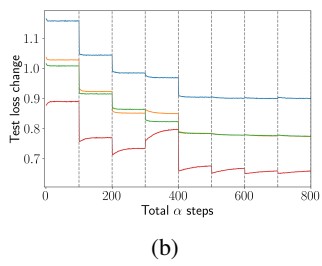 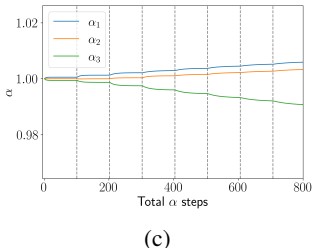

(a) (b) (c)

Figure 4: (a, b) Test loss trajectories of different datasets in the multi-episode setting, without quantized $\boldsymbol{\alpha}$ updates (c) the corresponding $\alpha$ values during Stage II. The vertical dashed lines separate different episodes.

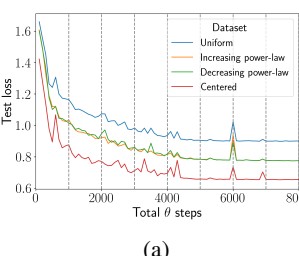 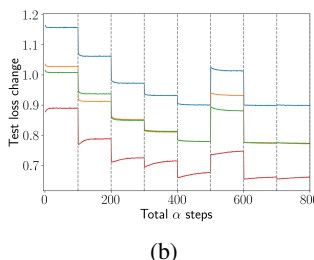 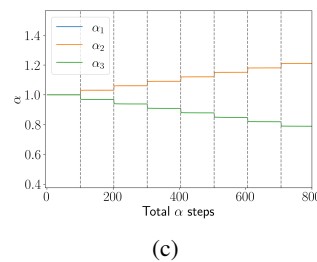

(a) (b) (c)

Figure 5: (a, b) Test loss trajectories of different datasets in the multi-episode setting, with quantized $\boldsymbol{\alpha}$ updates (c) the corresponding $\alpha$ values during Stage II. The vertical dashed lines separate different episodes. The mixture coefficients $\alpha_1$ and $\alpha_2$ overlap in this experiment.

trajectories without and with quantized updates applied to $\boldsymbol{\alpha}$. We observe that the $\boldsymbol{\alpha}$ values do not change appreciably in the non-quantized case. In comparison, the $\boldsymbol{\alpha}$ values change significantly in the quantized case and result in an optimal dataset mixture.

## C PROOF OF THEOREM THEOREM 1

*Proof.* Our goal is to approximate the gradient of the validation loss with respect to $\boldsymbol{\alpha}$

$$\nabla_\alpha V(\boldsymbol{\theta}_T) = \frac{\partial}{\partial \boldsymbol{\alpha}} V(\boldsymbol{\theta}_T). \tag{11}$$

Note that in the above equation, the trained model parameters $\boldsymbol{\theta}_T$ implicitly depend on $\boldsymbol{\alpha}$. To approximate this mixture gradient, we start with the chain rule, which expands Equation (11) to

$$\nabla_\alpha V(\boldsymbol{\theta}_T) = \frac{\partial V}{\partial \boldsymbol{\theta}_T} \frac{\partial \boldsymbol{\theta}_T}{\partial \boldsymbol{\alpha}}. \tag{12}$$

We now differentiate Equation (2) to with respect to $\boldsymbol{\alpha}$ get

$$
\begin{aligned}
\frac{\partial \boldsymbol{\theta}_t}{\partial \boldsymbol{\alpha}} &= \frac{\partial \boldsymbol{\theta}_{t-1}}{\partial \boldsymbol{\alpha}} - \eta_{t-1} \frac{\partial}{\partial \boldsymbol{\alpha}} L'(\boldsymbol{\theta}_{t-1}; \boldsymbol{\alpha}) \\
&= \frac{\partial \boldsymbol{\theta}_{t-1}}{\partial \boldsymbol{\alpha}} - \eta_{t-1} \frac{\partial}{\partial \boldsymbol{\alpha}} \left[ \boldsymbol{\alpha} \cdot L'(\boldsymbol{\theta}_{t-1}) \right] \\
&= \frac{\partial \boldsymbol{\theta}_{t-1}}{\partial \boldsymbol{\alpha}} - \eta_{t-1} L'(\boldsymbol{\theta}_{t-1}) - \eta_{t-1} \boldsymbol{\alpha} \cdot \frac{\partial}{\partial \boldsymbol{\alpha}} \left[ L'(\boldsymbol{\theta}_{t-1}) \right].
\end{aligned} \tag{13}
$$

By induction (and noting that $\boldsymbol{\theta}_0$ is constant and has zero gradient), we have

$$\frac{\partial \boldsymbol{\theta}_T}{\partial \boldsymbol{\alpha}} = - \underbrace{\sum_{t=0}^{T-1} \eta_t L'(\boldsymbol{\theta}_t)}_{\text{first order terms}} - \boldsymbol{\alpha} \underbrace{\sum_{t=0}^{T-1} \eta_t \frac{\partial}{\partial \boldsymbol{\alpha}} \left[ L'(\boldsymbol{\theta}_t) \right]}_{\text{higher order terms}} \tag{14}$$

$$\frac{\partial \boldsymbol{\theta}_T}{\partial \boldsymbol{\alpha}} = -G - E. \tag{15}$$

where $E$ is an error term representing higher-order derivatives. Combining this with equation 12 gives us our result.

$$\frac{\partial}{\partial \boldsymbol{\alpha}} V(\boldsymbol{\theta}_T) = -V'(\boldsymbol{\theta}_T) \cdot (G + E) \tag{16}$$

$$\frac{\partial}{\partial \boldsymbol{\alpha}} V(\boldsymbol{\theta}_T) \approx -V'(\boldsymbol{\theta}_T) \cdot G. \tag{17}$$

$\square$

## D  NORMALIZED MIXTURE COEFFICIENT FORMULATION

In this section, we describe normalized $\boldsymbol{\alpha}$ formalism in which the mixture coefficients are normalized. Let $\phi(.)$ denote the normalizing function (for example, softmax), such that $\sum_{i=1}^{N} \phi_i(\boldsymbol{\alpha}) = 1$.

$$L(\boldsymbol{\theta}; \boldsymbol{\alpha}) = \sum_{i=1}^{N} \phi_i(\boldsymbol{\alpha}) L_i(\boldsymbol{\theta}) = \phi(\boldsymbol{\alpha}) \cdot L(\boldsymbol{\theta}). \tag{18}$$

Under this normalization, the total change in the model parameters after $T$ steps is given by

$$\boldsymbol{\theta}_T = \boldsymbol{\theta}_0 - \sum_{t=1}^{T-1} \eta_t \phi(\boldsymbol{\alpha}) \cdot L'(\boldsymbol{\theta}_t) = \boldsymbol{\theta}_0 - \phi(\boldsymbol{\alpha}) \cdot \tilde{G}. \tag{19}$$

The derivative the $\boldsymbol{\theta}_T$ wrt $\boldsymbol{\alpha}$ is given by

$$\frac{\partial \boldsymbol{\theta}_T}{\partial \boldsymbol{\alpha}} \approx - \sum_{t=1}^{T-1} \phi'(\boldsymbol{\alpha}) \odot \tilde{G}. \tag{20}$$

Finally, the derivative of the validation loss is given by

$$\frac{\partial}{\partial \boldsymbol{\alpha}} V(\boldsymbol{\theta}_T) \approx -V'(\boldsymbol{\theta}_T) \cdot (\phi(\boldsymbol{\alpha}) \odot \tilde{G}). \tag{21}$$

