# OpenReview forum: "Gradient-based Optimization of Dataset Mixtures"
_ICLR.cc/2025/Conference — ICLR 2025 Conference Withdrawn Submission_

### Official Review · Reviewer_crhT · 2024-10-17

**Soundness:** 2
**Presentation:** 2
**Contribution:** 2
**Rating:** 5
**Confidence:** 4

**Summary:**

The authors introduce a gradient-based optimization framework that directly minimizes the loss on downstream tasks by adjusting the mixture weights of different data subsets. Central to their approach is the approximation of the "data mixture gradient," which facilitates tractable optimization despite the complex dependencies between mixture coefficients and model parameters. The paper presents a two-stage optimization process:

- **Stage I:** Optimizing model parameters using initial mixture coefficients while accumulating gradient contributions from each data subset.
- **Stage II:** Optimizing mixture coefficients to improve validation performance by leveraging the accumulated gradients from Stage I.

The authors provide theoretical justification for their gradient approximation and demonstrate the effectiveness of their method through experiments.

**Strengths:**

This paper introduces an innovative approach to optimizing data mixture coefficients. The "data mixture gradient" and two-stage optimization offer a fresh perspective on enhancing downstream performance. The method’s theoretical foundation is strong, with clear justifications and proofs for the gradient approximation. Experimental evaluations across diverse datasets show the method’s versatility and robustness. The paper is well-organized, making complex ideas accessible.

**Weaknesses:**

First, the gradient approximation assumes a small, constant learning rate ($\eta$), yet the paper lacks evidence on its performance with varying strategies like adaptive or scheduled rates. This raises concerns about its use in dynamic environments. Second, while the softmax function normalizes mixture coefficients for interpretability, its impact on optimization and performance compared to unconstrained coefficients is not explored. An empirical comparison would clarify this. Additionally, the paper omits potential theoretical limitations of gradient-based mixture optimization, leaving unclear if specific data distributions or tasks could hinder its performance. Please see questions below.

**Questions:**

- **Gradient Approximation Validity:**
  Your theoretical justification for the gradient approximation relies on the assumption of a small, constant learning rate ($\eta$). Can you provide empirical evidence or further theoretical insights on how the approximation performs with other learning rate strategy?

- **Impact of Mixture Coefficient Constraints:**
  You mention using a softmax function to normalize mixture coefficients for interpretability, ensuring they sum to one. How does this constraint affect the optimization process and the final performance compared to allowing unconstrained mixture coefficients? Have you compared both approaches empirically?

- **Theoretical Limits of Gradient-Based Mixture Optimization:**
  Beyond the practical scenarios tested, are there theoretical limitations to your gradient-based mixture optimization approach? For instance, are there specific types of data distributions or downstream tasks where your method might underperform or fail to provide benefits?

---

### Official Review · Reviewer_FJQq · 2024-10-28

**Soundness:** 2
**Presentation:** 3
**Contribution:** 2
**Rating:** 3
**Confidence:** 4

**Summary:**

The paper introduces a gradient-based method to optimize the data mixture coefficients for various end tasks. Specifically, the author proposes a two-stage strategy to iteratively update model parameters and the data mixture weights. A first-order approximation is applied for efficient computation for the gradients on the mixture weights.
The authors experiment their method on both language domain (Dyck grammar) and vision domain (CIFAR10, COCO), where the two-stage updating strategy could effectively improve above stage-1 only in most of cases.

**Strengths:**

1. The paper is well-written and easy to follow;
2. The authors conducted experiments across both language and vision domains with ablation studies according to various optimizers and amount of validation samples.

**Weaknesses:**

1. **The experiments lack comparisons to several baseline methods on data mixture optimization: (i) DoReMi [1] (ii) DOGE [2] (iii) The clustering-based method [3],** where the data mixture coefficient $\alpha$ can be obtained by assigning all validation samples into training datasets. [2] should be considered closely relevant since it also applies gradient-based method in data mixture weights optimization with similar first-order approximation. Also, the uniform distribution baseline is not considered on the language task (dyck grammar). In the vision experiments, the baseline methods included in the paper need a more clear clarification (e.g. the `Exact` method in Table2 and `Base` in Table5).
2. **The results from the language experiment (dyck grammar) is not clearly demonstrated.** e.g. what is the test perplexity compared to uniform baseline (train with uniform $\alpha$ distribution)? and what does the "uniform" line stands for in Figure 4,5 (a,b) given there are only three training datasets considered?
3. **The improvement on the vision tasks seem to be trivial.** In Table 2, the proposed method is not able to outperform the `Exact` baseline in most of cases, and all listed results in Table 3.
4. The authors did not give detailed analysis on the computation overheads by applying the methods.

[1] DoReMi: Optimizing Data Mixtures Speeds Up Language Model Pretraining
[2] DoGE: Domain Reweighting with Generalization Estimation
[3] Automatic Data Curation for Self-Supervised Learning: A Clustering-Based Approach

**Questions:**

1. Methodology: In stage-2, would you only use old gradient from previous checkpoints as $G$? or you need to compute new gradients on $\theta_T$? It states in Sec. 3.3 "In Stage II, we sample new random training examples every 100 iteration and recompute Gi accordingly."
2. Experiment: How does the proposed method perform comparing to the existing data mixture optimization baselines? and how does it perform comparing to training with uniform distribution with the same number of steps from Stage-1 and Stage-2?
3. Experiment: what is the base dataset corresponding to $\alpha_1, \alpha_2$ for the imbalanced CIFAR and object detection experiments?
4. What's the computation and memory requirement for the proposed method? It seems the proposed would only bring some improvement above the baseline if the gradient is estimated by 10,000 examples (Table2), which could incur a considerable overhead.
5. The number of training datasets (resp. distributions) are relatively small ($|\alpha|=2,3$). How is the scalability of the proposed method to the increasing number of subsets (e.g. $|\alpha|=10$ for CIFAR10 and 100 for CIFAR100)?

---

### Official Review · Reviewer_AtQ6 · 2024-11-02

**Soundness:** 2
**Presentation:** 2
**Contribution:** 2
**Rating:** 5
**Confidence:** 2

**Summary:**

This paper introduces a method to improve model performance on specific downstream tasks by re-weighting the importance of different training datasets. The authors propose a two-stage approach based on some approximations: in the first stage, they train the model while tracking the gradient contributions from each dataset. In the second stage, they optimize the mixture weights to minimize the validation loss on the target task, using a gradient-based method with a theoretically justified approximation for efficiency. This approach is tested on diverse tasks,  showing improved task-specific performance by dynamically re-weighting dataset contributions. The authors also introduce extensions for handling large datasets and multi-episode refinement. Overall, the paper provides a novel and effective strategy for optimizing dataset mixtures to better align model training with downstream task objectives.

**Strengths:**

- The paper is well-motivated and the problem of fine-tuning a model on the heterogeneous datasets efficiently is important.
- The proposed method is novel in my opinion. It is conceptually simple and theoretically grounded. The authors also provide approaches for large scale training and multi-episode training, making the method more practical.
- The paper presents results across multiple domains and provides ablation studies for more insights about the method.

**Weaknesses:**

- **Writing and Notation**: The paper’s writing could be improved for clarity. For example, notation consistency should be addressed:
    - In Equation 5 and 7, the gradient term should use boldface as $\mathbf{G}$ instead of $G$ to maintain consistency with earlier notation.
    - Caption in Table 1, $\theta$-optimization should be boldface as well.
- **Concept Explanation**: Certain concepts could be explained more clearly to improve readability:
    - Lines 120-121 mention that $\mathbf{\theta}_T$​ implicitly depends on $\mathbf{\alpha}$, but it lacks an explanation. It would be helpful to elaborate on this a little bit.
    - Line 194 references a change of variables that could benefit from additional explanation, perhaps in the appendix, as it may not be immediately clear to readers.
- **Limited Checkpointing Experiment**: The checkpointing experiment only includes 5 and 10 checkpoints. Expanding this to additional checkpoints would better illustrate the effect of this extension.
- **Efficiency Comparison**: While the method can be viewed as a more efficient fine-tuning approach where dataset weights are optimized, no direct comparison with conventional fine-tuning in terms of runtime or computational cost is provided.
- **Lack of Real-World NLP Task**: Although the motivation of the paper stems from natural language processing (NLP) applications, only a toy example (Dyck Grammar) is presented. Including a real-world NLP task would better demonstrate the method’s effectiveness and relevance for NLP applications.

Typos:
- line 307, paris -> pairs
- line 402, Qunatized -> Quantized

**Questions:**

- For the checkpointing experiment, it’s surprising that the model with 5 checkpoints outperforms the one with 10. I would expect a greater number of checkpoints to yield a more accurate gradient approximation and thus better performance. Additionally, more random examples improve performance with 5 checkpoints but degrade it with 10. Could you provide an explanation for these observations?
- Could you provide results of multi-episode optimization with quantized updates for all values of $\gamma$, as well as the results for multi-episode optimization without quantized updates across benchmarks?
- What does "Base" mean in Table 5? What is the set up for this baseline?

---

### Official Review · Reviewer_neEB · 2024-11-02

**Soundness:** 2
**Presentation:** 2
**Contribution:** 1
**Rating:** 3
**Confidence:** 4

**Summary:**

This paper proposes a method to automatically determine the weighting factor for training data from different sources.  The proposed method makes use of the validation loss and optimizes it over the weight space, also by gradient descent.  Empirical evidences are provided to demonstrate the success of the proposed method.

**Strengths:**

The general idea is valid.

**Weaknesses:**

The idea of this paper is simple and obvious. The working of the proposed approach depends on how similar the testing distribution is to the validation distribution and the size of the validation set.  Clearly, if the testing distribution is far from the data distribution from which the validation set is drawn, then optimization of the validation loss (on the validation set) would not lead to good testing performance. On the other hand, if the validation distribution is identical to the testing distribution, but the validation set is too small, this approach will overfit to the validation set, and also fail to generalize to the testing set.  To this reviewer, theoretically understanding the interplay between distribution shifts (from training, to validation, to testing) and the sizes of the training set and testing set is an important aspect of this research, but is ignored by the authors.

The proposed approach makes use the validation set. Then a natural question to be answered is this: among all approaches having access to both the training set and the validation set, is the proposed approach the best?

Overall, I feel the research results presented so far are shallow and lack nutrient.

**Questions:**

See weakness

---

### Official Review · Reviewer_KDrY · 2024-11-04

**Soundness:** 2
**Presentation:** 3
**Contribution:** 2
**Rating:** 5
**Confidence:** 4

**Summary:**

The paper investigates model training when the dataset is made of multiple different data sources, focusing on finding the best weight for each of these. The proposed approach tackles the task in a two-stage fashion, first accumulating the gradients for each data source during a standard training phase, and then optimizing for the coefficients of the linear combination of these gradients that minimizes a downstream validation loss. The work further introduces a checkpointing variant to mitigate the computational burden of the approach when facing large-scale datasets by only saving an equally spanned subsequence of the gradients, as well as a multi-episode version that repeats the two-stage process several times to achieve some additional gains. A carefully crafted experiment on a formal language intuitively confirms the behavior of the approach, while additional experiments on image classification on ad hoc versions of CIFAR10 and object detection on COCO validate its effectiveness.

**Strengths:**

- To the best of my knowledge, the presented approach is novel. The problem it tackles is well-defined and reasonable, as modern realistic datasets are always constituted by a suite of different data sources.
- The approach is extremely simple and general: by making no restrictive assumptions, the approach is model-agnostic and not tied to any modality or task. Any gradient-based learning pipeline can leverage it.
- The paper is well-written and clear, making it easy to read. The experiments are thoroughly described and the mathematical intuition is sound.

**Weaknesses:**

- My main concern is that the experimental evaluation is mostly made of toy/invented problems. Given the generality of the broader problem, I would have expected to see at least one real instance where the approach shines. The only real case in the paper is MS-COCO, but I am having a hard time understanding why it is a study case in the first case. What are the different data sources in MS-COCO? The introduction states “The Pile” as a case, being made of data from ArXiv, GitHub, etc. Can an experiment be made on this one instead?
- Lack of real baselines: the paper lists 6 previous works that investigated the optimization process of data mixture coefficients, yet does not compare with any of these. Having virtually no baseline tackling the same problem, it is hard to assess the virtues of the approach.
- The constant $\mathbf{G}$ approximation is not empirically validated. It would be helpful to see some quantitative measure assessing how much it actually holds, and whether it correlates with the benefits of the approach.

**Questions:**

- I feel like the work is somewhat related to Task Arithmetics [1], where the final multi-task model is obtained from a linear combination of task-specific (here they would be data source-specific) vectors, constituted by differences between fine-tuned models on those tasks and the pretrained base. While task vectors already sort of resemble the $\mathbf{G}$s seen in this paper, they also optimize the combination coefficients over a validation set. It would be interesting to see some discussion on this aspect if the authors find it relevant.
- L195: “This converts Equation 6 (..)” shouldn’t this be Equation 4?

**Typos**:
- Equations are often preceded by a newline (e.g. equation 1, 6, 7), adding unnecessary space.
- 402: Qunatized
- 405: epsiode
- 418: citet → citep

[1] Ilharco, Gabriel, et al. "Editing models with task arithmetic." The Eleventh International Conference on Learning Representations.

---

### Note · Authors · 2024-11-15

I have read and agree with the venue's withdrawal policy on behalf of myself and my co-authors.